# Volumetric Imitation Generative Adversarial Networks for Anatomical Human Body Modeling

**DOI:** 10.3390/bioengineering11020163

**Published:** 2024-02-07

**Authors:** Jion Kim, Yan Li, Byeong-Seok Shin

**Affiliations:** Department of Electrical and Computer Engineering, Inha University, Incheon 22212, Republic of Korea; 3161508@gmail.com (J.K.); leeyeon@inha.ac.kr (Y.L.)

**Keywords:** GAN, imitation, 3D reconstruction, volumetric representation, human body, deep learning

## Abstract

Volumetric representation is a technique used to express 3D objects in various fields, such as medical applications. On the other hand, tomography images for reconstructing volumetric data have limited utilization because they contain personal information. Existing GAN-based medical image generation techniques can produce virtual tomographic images for volume reconstruction while preserving the patient’s privacy. Nevertheless, these images often do not consider vertical correlations between the adjacent slices, leading to erroneous results in 3D reconstruction. Furthermore, while volume generation techniques have been introduced, they often focus on surface modeling, making it challenging to represent the internal anatomical features accurately. This paper proposes volumetric imitation GAN (VI-GAN), which imitates a human anatomical model to generate volumetric data. The primary goal of this model is to capture the attributes and 3D structure, including the external shape, internal slices, and the relationship between the vertical slices of the human anatomical model. The proposed network consists of a generator for feature extraction and up-sampling based on a 3D U-Net and ResNet structure and a 3D-convolution-based LFFB (local feature fusion block). In addition, a discriminator utilizes 3D convolution to evaluate the authenticity of the generated volume compared to the ground truth. VI-GAN also devises reconstruction loss, including feature and similarity losses, to converge the generated volumetric data into a human anatomical model. In this experiment, the CT data of 234 people were used to assess the reliability of the results. When using volume evaluation metrics to measure similarity, VI-GAN generated a volume that realistically represented the human anatomical model compared to existing volume generation methods.

## 1. Introduction

Volumetric representation [1,2] is a popular technique to express 3D objects, such as surface modeling [3,4]. Volumetric data are generated mainly by reconstructing from tomographic images, such as computed tomography (CT) and magnetic resonance imaging (MRI) [5,6]. On the other hand, it is challenging to adopt tomographic images for purposes other than medical, such as diagnosis and surgery, because they contain personal information. GAN-based medical image generation techniques [7,8] can produce anatomically meaningful virtual tomographic images applicable to volume reconstruction. On the other hand, the image generation process does not consider the relationship between the adjacent slices of a volume because these methods account for 2D correlations exclusively. This can lead to erroneous results when reconstructing volumes from generated images, particularly when maintaining the 3D structural coherence between adjacent slices. While techniques have been proposed to generate 3D volumes [9,10], these approaches have been limited to generating surface representations and have failed to capture the internal characteristics. Consequently, a volume generation method is needed to reflect the entire 3D human anatomical model, including its internal portion.

This paper introduces the volumetric imitation GAN (VI-GAN), a novel approach that aims to imitate human anatomical models to generate volumetric data. The primary goal of this approach is to generate 3D models that faithfully capture the attributes and 3D structure (external shape, internal slice, and the relationship between vertical slices) within the human anatomical model. The proposed network comprises two main components: a generator to obtain the volumetric data and a discriminator to evaluate the authenticity between the generated volume and the ground truth. The generator performs feature extraction and up-sampling to produce a volume based on the 3D U-Net [11] and ResNet [12] structures. Moreover, the initial feature extraction process from the input image set uses a 3D-convolution-based LFFB (local feature fusion block) [13] to incorporate features at various scales. On the other hand, the discriminator uses 3D convolution to extract the authenticity of the generated volume and ground truth. Using the proposed network structure makes it possible to account for vertical correlations in the volume generation process compared to existing 2D image generation techniques. Moreover, it provides a more realistic representation compared to previous methods focused solely on surface generation because it can faithfully intimate the internal features of the anatomical model. The volume comprises 3D data with higher dimensions than the image, so converging in a specific shape is difficult. Therefore, if the basic distance loss alone is applied to generate the volume, it barely converges in the form of a human anatomical model [14]. Thus, the proposed method devises a reconstruction loss so VI-GAN can generate the volume converging to the human anatomical model. Reconstruction loss includes feature loss and similarity loss. Feature loss is calculated using an overlapping region between the generated volume and the ground truth. Similarity loss is also calculated as an internal similarity based on a structural similarity index map (SSIM) [15].

The spine data from the Digital Korean dataset [16] provided by the Korean Institute of Science and Technology Information (KISTI) and the liver data from CT volumes with multiple organ segmentations (CT-ORG) [17] were applied during the experiment to validate the proposed technique. Evaluation metrics, such as F1-score, dice coefficient, peak signal-to-noise ratio (PSNR), and universal image quality index (UQI), were used to measure the resemblance between the generated volume and the human anatomical model. The VI-GAN outperformed existing methods by producing volumes closely representing the human anatomical model.

The volumetric data generated by the VI-GAN included the external shape and internal structure of the human anatomical model. Therefore, it can be used in various fields, such as diagnosis [18,19] and surgical simulation [20,21]. VI-GAN can produce a 3D human anatomical model that can enhance training efficiency and immersion for medical professionals. Moreover, virtual tomographic images can be produced by decomposing the volumetric data generated by the VI-GAN. Compared to existing medical image generation methods, these images have fewer errors in the relationship between neighboring slices. The tomographic images produced by the VI-GAN can enhance the capabilities of medical professionals to distinguish diseases effectively during the diagnostic process.

The contributions of this paper are as follows. (1) This paper proposes a VI-GAN to create 3D volumetric data that capture the attributes and 3D structure of a human anatomical model (external shape, internal slice, and vertical slice relationships). (2) This paper introduces reconstruction loss that encompasses feature loss and similarity loss to enhance the convergence rate of VI-GAN in volume generation. The feature loss measures the overlapping regions, while the similarity loss quantifies the resemblance between the generated volume and ground truth.

## 2. Related Works

Several studies have proposed generating the surface of a human anatomical model in the volume format from tomographic images. Balashova et al. [9] proposed a method to reconstruct the surface of the liver using a single X-ray image. They used mask data and images in the training process to generate liver data closer to the ground truth. Henzler et al. [22] generated the surface of animal bones using multiple X-ray images. Their study combined the volumes generated from images of multiple viewpoints. It allowed the production of a high-quality volume with a complete reconstruction of the parts that could be obscured easily from a single viewpoint. Kasten et al. [10] devised a network to reconstruct knee bones from bi-planar X-ray images. Their study synthesized the volumes produced by duplicating axial, coronal, and sagittal images in the z-axis direction in the training process. On the other hand, the generated data could not preserve the characteristics of the entire portion of that model because these methods generated only the surface of the human anatomical model. Furthermore, many of these studies have been proposed in the form of CNNs. The proposed technique requires generating virtual volume data. Hence, applying a GAN specialized for generation is necessary.

Several studies have proposed producing 2D medical data using GANs. These studies have focused mainly on the synthesis and reconstruction of images. Synthesis techniques include changing the style and modality [23,24] and adding characteristics, such as nodules and tumors [25]. Reconstruction techniques cover the super-resolution [26] process. Among these techniques, medical image generation [7,27] can produce the virtual tomographic images required for volume reconstruction. Chuquicusma et al. [28] and Frid-Adar et al. [29] proposed techniques for generating images representing lung modules and liver lesions using a deep convolution GAN (DCGAN). Beer et al. [8] devised a method for generating tomographic images through the progressive growing of GAN (PGGAN) [30] to express skin lesions realistically. These methods generated tomographic images to improve the classification and segmentation performance during the training process. Nevertheless, these medical image generation techniques considered only the 2D correlation within the image. Therefore, erroneous results can be obtained when reconstructing volumes from these generated images because the vertical correlation with the adjacent slices is not considered. Hence, the volume generation technique using a GAN should be proposed to prevent such erroneous results.

Some studies adopted the GAN structure for volume generation by expanding various image generation techniques into 3D space. Wu et al. [31] proposed 3D-GAN and 3D variational autoencoder GAN (3D-VAE-GAN) structures to produce the surface of an object as a volume using a single 2D image. Smith et al. [32] applied the Wasserstein distance to 3D-GAN structures, which improved the volume quality. Volume-based GANs are applied in the medical field, such as classification [33], segmentation [34], denoising [35], and detection [36]. Vox2Vox [37] is one of the volume-based GAN techniques used in the medical field for segmenting brain tumors. Nevertheless, few studies have applied GAN structures in volume generation, particularly for human anatomical models. Thus, it is essential to devise a method for generating volume data similar to a human anatomical model using a GAN.

## 3. Methods

### 3.1. Training Process

Figure 1 presents the overall network structure and components of the VI-GAN. The VI-GAN aims to generate volumetric data similar to the human anatomical model. This generation process is achieved using the volume generator G, which incorporates a 3D-convolution-based LFFB (local feature fusion block) [13], denoted as the 3D LFFB. The discriminator D is used to assess the authenticity of the generated volume and the ground truth. In addition, the VI-GAN devises reconstruction loss, including distance, feature, and similarity losses, to converge the generated volumes as a human anatomical model.

The proposed method should reconstruct all the voxels in the generated volume similar to the corresponding voxels in the ground truth volume. The difference between any two corresponding values should be close to zero. Equation (Equation 1) represents the training objective for a volumetric dataset. V represents the ground truth volume given during training, and $\hat{V}$ denotes the corresponding generated volume. V[m] and $\hat{V}\left[m\right]$ are the mth voxel of the ground truth volume V and generated volume $\hat{V}$, respectively. l, w, and h represent x, y, and z-axis resolutions of the volume, respectively.
(1)$$\begin{array}{ccc} \; & \forall m\left|V\left[m\right]-\hat{V}\left[m\right]\right|\approx 0 & \\ \; & \;0\leq m\leq l\times w\times h,\forall m(V\left[m\right],\hat{V}\left[m\right]\in R^{l\times w\times h}) \end{array}$$

Volume generator G produces a volume using the input image set $I^{G}$. The structure of G is represented in Figure 1b using Equation (Equation 2). $X_{k}$ represents the intermediate result in the kth layer. The input image set is reconstructed into a volume via each f layer. In the overall volume generation process, the network structures based on 3D U-Net [11] and ResNet [12] are categorized into the encoding, refining, and decoding parts. The encoding, refining, and decoding parts comprise $f_{k}^{\mathrm{down}}$, $f_{k}^{\mathrm{mid}}$, and $f_{k}^{\mathrm{up}}$ layers, respectively. The final layer is formed as the $f^{\prime }$ layer; k is the index of those layers. $\theta_{k}^{G}$ is the learning parameter of the kth layer in the generator. Before passing to the f layers, 3D LFFB is performed to extract the essential features of the image set. Figure 1d presents the network structure of 3D LFFB.

$f_{k}^{\mathrm{down}}$ is the convolution layer that extracts the features from the input through down-sampling. $f_{k}^{\mathrm{mid}}$ is the convolution layer that refines features using the ResNet architecture. $f_{k}^{\mathrm{up}}$ is the deconvolution layer that reconstructs a volume from features by up-sampling. The final layer $f^{\prime }$ is a convolution layer that generates a volume $\hat{V}$ in which all voxel values are normalized from 0.0 to 1.0. The kernel sizes of all layers are 4×4×4, and the stride sets were assigned for $f_{k}^{\mathrm{down}}$, $f_{k}^{\mathrm{up}}$, and $f^{\prime }$ as two and $f_{k}^{\mathrm{mid}}$ as one. $n_{G}$ indicates the total number of layers of the generator; d, r, and u are the layer indices of the encoding, refining, and decoding parts, respectively.
(2)$$\begin{array}{ccc} \; & X_{0}=I^{G},X_{d}=f_{d-1}^{\mathrm{down}}\left(X_{d-1};\theta_{d-1}^{G}\right),\;X_{r}=f_{r-1}^{\mathrm{mid}}\left(X_{r-1};\theta_{r-1}^{G}\right), & \\ \; & X_{u}=f_{u-1}^{\mathrm{up}}\left(X_{u-1};\theta_{u-1}^{G}\right)\;\hat{V}=f^{\prime }\left(X_{n_{G}-1};\theta_{n_{G}-1}^{G}\right), & \\ \; & 0<d<r<u<n_{G} \end{array}$$

Volume discriminator D distinguishes the authenticity between the generated and ground truth volume. The structure of D is represented in Figure 1c with Equation (Equation 3). $I^{D}$ is the input volume. $Y_{k}$ represents the intermediate result in the kth layer. The probability p indicating the authenticity of the input volume is calculated through each g layer. $\theta_{k}^{D}$ is the learning parameter of the kth layer in the discriminator. $g_{k}$ is a convolution layer that extracts the features from $I^{D}$ by down-sampling. The final layer, $g^{\prime }$, is a fully connected layer to generate a probability normalized from 0.0 to 1.0. The kernel sizes and stride sets of all layers are 4×4×4 and 2, respectively. $n_{D}$ represents the total number of layers in the discriminator.
(3)$$\begin{array}{ccc} \; & Y_{0}=I^{D},\;Y_{k}=g_{k-1}\left(Y_{k-1};\theta_{k-1}^{D}\right), & \\ \; & p=g^{\prime }\left(Y_{n_{D}-1};\theta_{n_{D}-1}^{D}\right),\;0<k<n_{D} \end{array}$$

### 3.2. Loss Function

The loss function of the generator and discriminator was designed, as shown in Equation (Equation 4). $L_{G}$ and $L_{D}$ are the generator and discriminator loss, respectively, of the ground truth volumes V, generated volumes $\hat{V}$, and the input image set of the generator $I^{G}$. $L_{\mathrm{recon}}$ is the reconstruction loss, and α is a constant weight assigned to $L_{\mathrm{recon}}$.
(4)$$\begin{array}{ccc} \; & L_{G}=E_{x∼I^{G},\hat{v}∼\hat{V}}{∥D(x,\hat{v})-1∥}_{2}^{2}+\alpha L_{\mathrm{recon}} & \\ \; & L_{D}=E_{x∼I^{G},v∼V,\hat{v}∼\hat{V}}\left[{∥D(x,v)-1∥}_{2}^{2}+{∥D(x,\hat{v})∥}_{2}^{2}\right] \end{array}$$

The reconstruction loss $L_{\mathrm{recon}}$ calculates the discrepancy between the generated and the ground truth volumes. Reconstruction loss consists of the distance, feature, and similarity loss. Among these losses, the distance loss $L_{\mathrm{dist}}$ is defined using Equation (Equation 5). The L1 loss is used for the distance loss.
(5)$$L_{\mathrm{dist}}=E_{v∼V,\hat{v}∼\hat{V}}{∥v-\hat{v}∥}_{1}$$

The voxel values in the generated volume indicate the predicted density of the corresponding area of the human anatomical model. The ground truth volume contains many undefined parts outside the human anatomical model, represented by low-value voxels. Therefore, many voxels in the ground truth volume have low values. The proposed method should reconstruct the entire part of the human anatomical model. Nevertheless, when only the distance loss is used in the regularization term, all voxels are averaged to minimize the distance loss. This process reduces the value of the high-value voxels [14]. The generated volume barely converges with the target model because the shape and characteristics of the human anatomical model are composed mainly of a high-value area. This paper proposes a reconstruction loss that consists of distance loss with feature loss and similarity loss to solve this problem.

Feature loss represents how many high-value voxels overlap between two volumes. Feature loss can be used to emphasize and preferentially reconstruct high-value voxels during the volume-reconstruction process. Feature loss $L_{f}$ is expressed as Equation (Equation 6). This loss uses $N_{r}$ number of thresholds; $t_{s}$ is the sth threshold; m is the voxel index in the volume; l, w, and h are the x, y, and z-axis resolutions of the volume, respectively. I(·) is the indicator function.
(6)$$\begin{array}{ccc} \; & L_{f}=1-E_{v∼V,\hat{v}∼\hat{V}}\left[\sum_{s}^{N_{r}}\frac{\sum_{m}^{l\times w\times h}I\left(v\left[m\right]\geq t_{s}\right)I\left(\hat{v}\left[m\right]\geq t_{s}\right)}{\sum_{m}^{l\times w\times h}I\left(I\left(v\left[m\right]\geq t_{s}\right)+I\left(\hat{v}\left[m\right]\geq t_{s}\right)\right)}\right] & \\ \; & 0\leq t_{s}\leq 1 \end{array}$$

Feature loss can be used to generate the characteristics of a human anatomical model, composed mainly of drastic changes in density. Such characteristics are often represented primarily by a high value. The high-value area within the human anatomical model can be fully reconstructed in the output volume by assigning feature loss during the training process.

Similarity loss $L_{\mathrm{sim}}$ reflects the difference in image quality of each internal image slice between the generated and ground truth volumes, which is defined in Equation (Equation 7). The SSIM, SSIM(·,·) [15], measures the difference in image quality. Similarity loss computes the difference in volume quality calculated based on the SSIM between the internal image slices in the generated and ground truth volume for all z-values. S(·,k) is the selector that extracts the kth slice in the volume, and h is the resolution in the z-axis direction of the volume. Using the similarity loss, the internal voxel values can be reconstructed like those of the ground truth volume.
(7)$$L_{\mathrm{sim}}=1-E_{v∼V,\hat{v}∼\hat{V}}\left\{\frac{1}{h}\sum_{k=0}^{h}SSIM\left(S\left(v,k\right),S\left(\hat{v},k\right)\right)\right\}$$

The final reconstruction loss is represented as Equation (Equation 8).
(8)$$L_{\mathrm{recon}}=L_{\mathrm{dist}}+L_{f}+L_{\mathrm{sim}}$$

### 3.3. Experimental Setting

This study used CT images of the spine (fifth lumbar vertebra and right hip bone) from the Digital Korean dataset provided by the KISTI [16] and the liver from CT-ORG [17]. The CT data for 94 people for the spine and 140 people for the liver were used. Among them, the CT data of the following were applied: 70% for training, 20% for testing, and 10% for validation. A total of 3117 slices for the fifth lumbar vertebra (average of 33 slices per person), 4579 slices for the right hip bone (average of 49 slices per person), and 19,314 slices for the liver (average of 483 slices per person) were used to generate volumetric data for training. NVIDIA GeForce RTX 3090 Ti with 24,268 MB GPU memory was applied for training. The Adam optimizer [38] was used with a learning rate of $2\times 10^{-4}$. The dropout rates of the middle-ware network blocks in Figure 1b were set to 0.2. The constant α was set to 33.0 when implementing Equation (Equation 4).

## 4. Results

The generated volumes were evaluated using a confusion matrix [39]. Each voxel in the generated volume was classified as positive if it had a high value and negative if it had a low value. In addition, each voxel in the ground truth volume was also categorized as true if it had a high value and false if it had a low value. The states of the voxels were judged by a comparison with the threshold value, whether each voxel value was high or low. Equation (Equation 9) expresses the TP (true positive), FP (false positive), and FN (false negative) used for the evaluation metric. t is the threshold; v is the ground truth volume; $\hat{v}$ is the corresponding generated volume; l,w, and h are the x,y, and z-axis resolutions of the volume, respectively. The thresholds were used to evaluate how well the voxels that formulate a shape and internal characteristics of a human anatomical model were reconstructed. TP, FP, and FN were used to calculate the precision, recall, F1-score, and Dice coefficient [40].
(9)$$\begin{array}{ccc} \; & \mathrm{TP}=\frac{1}{l\times w\times h}\sum_{m=0}^{l\times w\times h}I\left(v\left[m\right]>t\right)I\left(\hat{v}\left[m\right]>t\right) & \\ \; & \mathrm{FP}=\frac{1}{l\times w\times h}\sum_{m=0}^{l\times w\times h}I\left(v\left[m\right]\leq t\right)I\left(\hat{v}\left[m\right]>t\right) & \\ \; & \mathrm{FN}=\frac{1}{l\times w\times h}\sum_{m=0}^{l\times w\times h}I\left(v\left[m\right]\leq t\right)I\left(\hat{v}\left[m\right]\leq t\right) \end{array}$$

Figure 2 compares the generated volumes between the proposed VI-GAN and existing methods. The quality of the generated volume was evaluated using three criteria: volumetric shape, thresholding result, and internal image slice. The volumetric shape describes how accurately the generated volume represents the external shape of the ground truth volume. The thresholding result describes the internal structure that represents how much high-value voxels overlap between generated and ground truth volume. The internal image slice expresses how similar the internal area is between the generated and ground truth volume. The value of each voxel is a floating point between 0.0 and 1.0. The positions with a lower or higher voxel value are blue or red, respectively. The thresholds were determined for thresholding results by analyzing the voxel value distributions within the ground truth volumes. Specifically, thresholds of 0.4, 0.3, and 0.28 were applied to the fifth lumbar vertebra, right hip bone, and liver, respectively. For the fifth lumber vertebra and right hip bone, the Q3 (third quantile, 75% of data points) [41] values were adopted as thresholds to emphasize the rigid areas within the skeletal system. For the liver, the Q1 (first quantile, 25% of data points) values were used as thresholds to visualize the soft tissue density. For the internal image slice, the slice position is the center of the volume. The slices correspond to the xy plane exactly, the xy plane leaning at $-45^{\circ }$, and the yz plane in the cases of the fifth lumbar vertebra, right hip bone, and liver, respectively. All slices in the CT volume were used as input data in Pix2Vox. The real CT volume was applied as input for the end-to-end CNN instead of the synthesized volume.

In the Pix2Vox and end-to-end CNN model, the volumetric shape had an ambiguous form that cannot express the human anatomical model in detail. In addition, the high-value voxels were scattered and could not typically converge to the shape of the human anatomical model. The generated volumes of the Vox2Vox model converged more to the human anatomical model than those of the Pix2Vox and end-to-end CNN models. The volumetric shapes and internal slices depicted as the form of the liver and bone represent this convergence. In the case of the fifth lumber vertebra and right hip bone, however, the high-value voxels in some areas of the volume were not fully reconstructed, which could not make the shape of the human anatomical model clear. In the case of the liver, the volumetric shape did not resemble the ground truth volume, and the organ shape did not appear in the center, which is the correct position for the internal slice. The VI-GAN generated a volumetric shape, high-value voxels, and internal parts that were more similar to the ground truth volume than the other methods. In conclusion, in a qualitative comparison, the VI-GAN generated volume data that are more similar to the human anatomical model than other existing methods.

Table 1 lists the quality measurement of the generated volume between the proposed VI-GAN and other existing methods. The quality was calculated using evaluation factors to measure the disparity between the generated and ground truth volume. The intersection over union (IoU), F1-score (F1), and Dice coefficient (DC) [43] represent the rate of overlapping voxels between thresholded results. The threshold values for calculating the IoU, F1, and DC metrics were 2.5, 1.9, and 2.8, respectively, corresponding to the fifth lumbar vertebra, right hip bone, and liver. These values represent the Q1 (25% of data points) value of the voxel distribution. The L1 error (L1) describes the voxel-wise difference between volumes. The peak signal-to-noise ratio (PSNR) [15], universal quality index (UQI) [44], visual saliency-induced index (VSI) [45], and structural similarity index map (SSIM) showed the similarity between the generated and ground truth volumes calculated using all slices.

A comparison of the VI-GAN with Pix2Vox and end-to-end CNN revealed that the VI-GAN had the highest result for all evaluation factors in all cases (the fifth lumber vertebra, right hip bone, and liver). As a result, the proposed method produces a volume that closely resembles the ground truth, displaying a higher rate of overlap among high-value voxels compared to Pix2Vox and end-to-end CNN. A comparison of the VI-GAN with Vox2Vox using the IoU, F1-score, and Dice coefficient showed that the VI-GAN produced better results in all cases than Vox2Vox. When comparing the metrics of L1, PSNR, UQI, VSI, and SSIM, it is difficult to definitively conclude whether the VI-GAN or Vox2Vox exhibited superior performance on similarity. Furthermore, the disparities in results were relatively minor in most cases. Based on these findings, the volumes produced by Vox2Vox and the VI-GAN showed similar degrees of resemblance to the ground truth. In summary, as listed in Table 1, the VI-GAN effectively generated a volume by accurately capturing high-value voxels compared to other methods while preserving the similarity to the human anatomical model.

Figure 3 presents the assessment results using the intersection over union, F1-score, and Dice coefficient across various threshold values. The objective of this experiment was to evaluate the accuracy in reconstructing high-value voxels, which holds significance in accurately representing the essential information within the volume. This information encompasses the overall appearance of soft tissue or rigid structures within the skeletal system. Furthermore, achieving precise reconstructions of high-value voxels is challenging because of the limited occurrence of such areas in the ground truth volume. The thresholds were selected within the range of Q1 (25% of data points) to Q3 (75% of data points), corresponding to 25%, 37.5%, 50%, 62.5%, and 75% within the voxel distribution. In this experiment, the following thresholds were applied: 0.2500, 0.2875, 0.3250, 0.3625, and 0.4000 for the fifth lumbar vertebra; 0.1900, 0.2175, 0.2450, 0.2725, and 0.3000 for the right hip bone; and 0.2800, 0.3250, 0.3700, 0.4150, and 0.4600 for the liver. Figure 3 shows a consistent trend across all methods in most cases; the results of the evaluation factor tended to decrease as the threshold value increased. This trend shows the difficulty of reconstructing high-value voxels within the volume accurately. Furthermore, the proposed VI-GAN consistently achieved superior results in most cases. Consequently, the VI-GAN demonstrated higher performance in reconstructing high-value voxels than the existing methods.

## 5. Conclusions

This paper proposed a VI-GAN for generating a volumetric model to describe the human anatomical model using a GAN-based volume generator and discriminator with 3D LFFB. This paper also proposed reconstruction loss, including feature loss and similarity loss, to reconstruct high-value areas and describe the essential characteristics of the model accurately. The experimental result showed that the generated volume of the VI-GAN represents the shape, high-value areas, and internal part better than the other existing methods evaluated. Furthermore, the experimental results with varying threshold values showed that the VI-GAN accurately generates high-value areas compared to existing methods. Based on these results, VI-GANs may enhance the availability of medical data and improve the training efficiency of medical professionals by generating high-quality volumetric data.

## Figures and Tables

**Figure 1 bioengineering-11-00163-f001:**
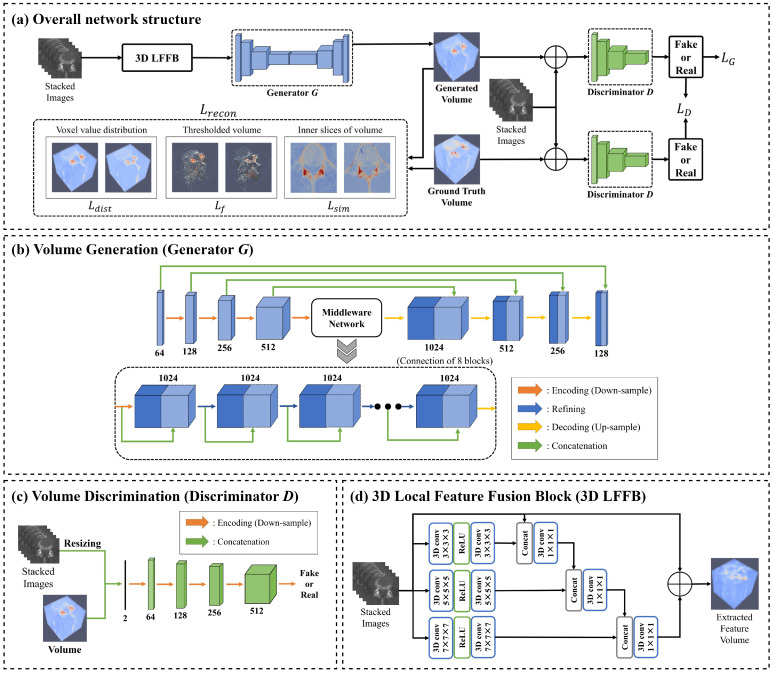
Overall proposed framework and components of VI-GAN. The blue cubes represent features extracted by the generator, while the green cubes depict features extracted by the discriminator. (**a**) Overview of the proposed network structure for generating a volume. (**b**,**c**) Detailed structure of the volume generator and discriminator. Three dots means repeating the preceding refining and concatenation process in the same manner. (**d**) 3D local feature fusion block (3D LFFB) that initially extracts the features from an image set before the volume generator.

**Figure 2 bioengineering-11-00163-f002:**
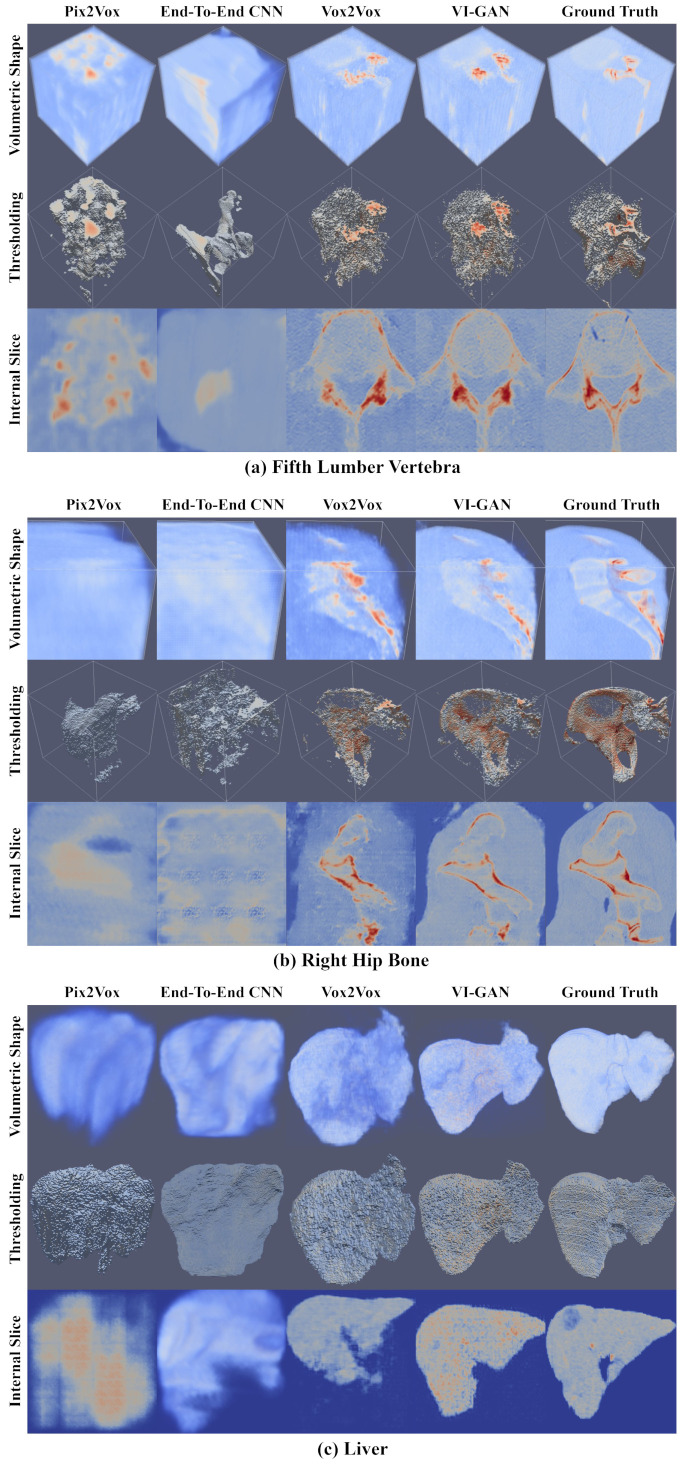
Qualitative comparison of the generated volumes in the proposed VI-GAN and existing method. The (**a**) fifth lumber vertebra, (**b**) right hip bone, and (**c**) liver volume data are represented. The images consist of a volumetric shape (top row), thresholding result (center row), and internal image slice (bottom row). The volumes generated by the Pix2Vox [42] (first column), end-to-end CNN [10] (second column), Vox2Vox [37] (third column), and VI-GAN (fourth column) models were compared.

**Figure 3 bioengineering-11-00163-f003:**
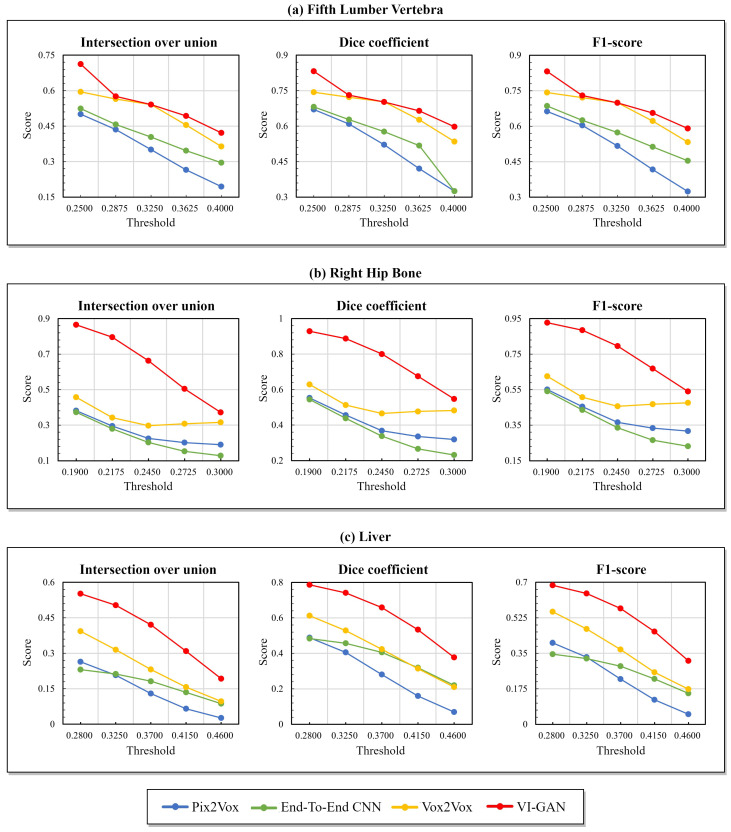
Quality comparison between the volumes of the proposed VI-GAN and existing methods across a range of thresholds.

**Table 1 bioengineering-11-00163-t001:** Quality comparison between the volumes of proposed VI-GAN and existing methods using the evaluation factors. The best results are represented in bold.

Method	Fifth Lumber Vertebra				Right Hip Bone				Liver			
	**IoU**	**DC**	**F1**	**L1**	**IoU**	**DC**	**F1**	**L1**	**IoU**	**DC**	**F1**	**L1**
Pix2Vox	0.501	0.670	0.662	0.121	0.381	0.553	0.551	0.131	0.264	0.488	0.402	0.139
End-to-end CNN	0.524	0.681	0.686	0.121	0.372	0.544	0.541	0.114	0.231	0.484	0.346	0.155
Vox2Vox	0.596	0.743	0.742	**0.096**	0.457	0.628	0.626	**0.075**	0.393	0.612	0.555	**0.053**
**VI-GAN**	**0.712**	**0.832**	**0.831**	0.102	**0.865**	**0.929**	**0.927**	**0.075**	**0.552**	**0.741**	**0.685**	0.056
**Method**	**Fifth Lumber Vertebra**				**Right Hip Bone**				**Liver**			
	**PSNR**	**UQI**	**VSI**	**SSIM**	**PSNR**	**UQI**	**VSI**	**SSIM**	**PSNR**	**UQI**	**VSI**	**SSIM**
Pix2Vox	16.226	0.839	0.827	0.329	15.764	0.729	0.829	0.279	15.486	0.884	0.831	0.284
End-to-end CNN	16.573	0.849	0.836	0.244	16.676	**0.817**	**0.872**	**0.613**	14.804	0.839	0.776	0.409
Vox2Vox	**18.249**	0.876	**0.875**	0.483	**24.615**	**0.817**	**0.872**	**0.613**	20.942	**0.900**	**0.880**	0.652
**VI-GAN**	18.189	**0.883**	0.874	**0.505**	24.518	0.816	0.865	0.600	**26.661**	0.886	0.874	**0.669**

## Data Availability

The spine CT images from the Digital Korean dataset are not accessible to the public. Please contact the Korea Institute of Science & Technology Information (KISTI) to obtain the data. The liver CT images from CT-ORG are publicly accessible and are available at https://wiki.cancerimagingarchive.net/pages/viewpage.action?pageId=61080890 (accessed on 30 January 2024).

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
