# Peer review of "Volumetric Imitation Generative Adversarial Networks for Anatomical Human Body Modeling"

_bioengineering, 2024, doi:10.3390/bioengineering11020163_

Round 1

Reviewer 1 Report

Comments and Suggestions for Authors

Kindly improve the quality of figures particularly Figure 4.

Kindly highlight the novelty in your work.

Kindly revise English throughout the manuscript.

1. What specific gap in the field does the paper address?
2. What does it add to the subject area compared with other published material?

3. Please revise the abstract and introduction including point 1 and point 2. 4. What specific improvements should the authors consider regarding the methodology? What further controls should be considered? Comments on the Quality of English Language

Major Revisions Needed

Author Response

Answers to reviewer's comments:
Paper title: Volumetric Imitation GAN for Anatomical Human Body Modeling
Author: Jion Kim, Yan Li, Byeong-Seok Shin
I'd like to give great thanks to dear associate reviewer. I appreciate your useful comments and advice. The answers to review comments are written in bold characters.
My responses are edited in the attached file.

Reviewer 2 Report

Comments and Suggestions for Authors

The study focuses on the use of Volumetric Imitation GAN (VI-GAN) to create three-dimensional representations of the human body that closely resemble anatomical features.

Although the manuscript is well written, there is still room for improvement.

Introduction:

Page 2, line 46: The spine data 45 from Digital Korean provided by KISTI and the liver data from CT-ORG. Please write the full name of KISTI and CT-ORG before using the abbreviations.

Page 2, line 66-70: Section 2 introduces the technology for generating data on human anatomical models using deep learning, including GAN. Section 3 describes the VI-GAN framework for generating volumetric data representing the human anatomical model. Section 4 compares the performance between the VI-GAN and the existing methods. The conclusions are reported in Section 5.

Nothing in the text after that mentions any section. Is this paragraph necessary?

Related works:

Page 2, line 73-79: [18] proposed a method to reconstruct the surface of the liver using a single X-ray image. They used mask data and images in the training process to generate liver data closer to the ground truth. [19] generated the surface of animal bones using multiple X-ray images. This study combined volumes generated from images of multiple viewpoints. It allowed the production of a high-quality volume with a complete reconstruction of the parts that could easily be obscured from a single viewpoint. [20] devised a network to reconstruct knee bones using bi-planar X-ray images.

Please use the name of authors if cite in the beginning of the text such as Balashova et al [18] proposed a method to reconstruct the surface of the liver using a single X-ray image instead of using the number only. The same goes to Page 3, line 94, line 104, and line 106.

Materials and Methods:

This section does not mention any data used in this study.

Results:

Page 6, line 191-200: This study used CT images of the spine (fifth lumbar vertebra and right hip bone) from the Digital Korean data provided by KISTI [12] and the liver from CT-ORG [13]. The CT data for 94 people for the spine and 140 people for the liver were used. Among them, the CT data of the following were applied: 70% for training, 20% for testing, and 10% for validation. A total of 3,117 slices for the fifth lumbar vertebra (average of 33 slices per person), 4,579 slices for the right hip bone (average of 49 slices per person), and 19,314 slices for the liver (average of 483 slices per person) were used to generate volumetric data for training. NVIDIA GeForce RTX 3090 Ti with 24268MB GPU memory was used for training. The Adam optimizer [38] was used with a learning rate of 2 x 10-4. The dropout rates of the middle-ware network blocks in Figure 1 (b) were set to 0.2. While implementing Equation (4), the constant a was set to 33.0.

This is not the result. This paragraph should move to the Methods section.

Comments on the Quality of English Language

Minor editing is required.

Author Response

(The authors gave the same response as above.)

Reviewer 3 Report

Comments and Suggestions for Authors

It is a very interesting paper even if its content is extremely specialized. The problem addressed is that of the reconstruction of volumetric images using tomographic images. It is in fact possible to visualize internal organs and anatomical structures from tomographic images, in this case CT; this is definitely a useful tool to facilitate diagnosis. In this case the use of AI is almost mandatory, moreover the article contains an extensive bibliography on the topic. A new method is proposed which is based on Volumetric Imitation GAN. In summary, the volumetric image is obtained, as well described in figure 1, using a reduced number of tomographic images. The most significant part of the paper is the comparison with other techniques, which always use AI, proposed in the literature, using some quality indicators. The comparison provides positive results for almost all previously proposed techniques.

The summary is well written and the bibliography quite detailed and relevant. The images and tables are of good quality.

I certainly believe that the text can be published in its current form.

Comments on the Quality of English Language

English is understandable and well written

Author Response

(The authors gave the same response as above.)

Round 2

Reviewer 1 Report

Comments and Suggestions for Authors

Please Accept